# Nevirapine Biotransformation Insights: An Integrated In Vitro Approach Unveils the Biocompetence and *Glutathiolomic* Profile of a Human Hepatocyte-Like Cell 3D Model

**DOI:** 10.3390/ijms21113998

**Published:** 2020-06-03

**Authors:** Madalena Cipriano, Pedro F Pinheiro, Catarina O Sequeira, Joana S Rodrigues, Nuno G Oliveira, Alexandra M M Antunes, Matilde Castro, M Matilde Marques, Sofia A Pereira, Joana P Miranda

**Affiliations:** 1Fraunhofer Institute for Interfacial Engineering and Biotechnology IGB, 70569 Stuttgart, Germany; madalena.cipriano@igb.fraunhofer.de; 2Research Institute for Medicines (iMed.ULisboa), Faculty of Pharmacy, Universidade de Lisboa, 1649-003 Lisbon, Portugal; pfpinheiro@gmail.com (P.F.P.); joana.s.rodrigues@campus.ul.pt (J.S.R.); ngoliveira@ff.ulisboa.pt (N.G.O.); mcastro@ff.ul.pt (M.C.); 3Centro de Química Estrutural (CQE), Instituto Superior Técnico, Universidade de Lisboa, 1049-001 Lisbon, Portugal; alexandra.antunes@tecnico.ulisboa.pt (A.M.M.A.); matilde.marques@tecnico.ulisboa.pt (M.M.M.); 4Centro de Estudos de Doenças Crónicas (CEDOC), NOVA Medical School/Faculdade de Ciências Médicas, Universidade Nova de Lisboa, 1169-056 Lisbon, Portugal; catarina.sequeira@nms.unl.pt (C.O.S.); sofia.pereira@nms.unl.pt (S.A.P.)

**Keywords:** hepatocytes, nevirapine, stem cells, 3D culture, glutathione, metabolism

## Abstract

The need for competent in vitro liver models for toxicological assessment persists. The differentiation of stem cells into hepatocyte-like cells (HLC) has been adopted due to its human origin and availability. Our aim was to study the usefulness of an in vitro 3D model of mesenchymal stem cell-derived HLCs. 3D spheroids (3D-HLC) or monolayer (2D-HLC) cultures of HLCs were treated with the hepatotoxic drug nevirapine (NVP) for 3 and 10 days followed by analyses of Phase I and II metabolites, biotransformation enzymes and drug transporters involved in NVP disposition. To ascertain the toxic effects of NVP and its major metabolites, the changes in the glutathione net flux were also investigated. Phase I enzymes were induced in both systems yielding all known correspondent NVP metabolites. However, 3D-HLCs showed higher biocompetence in producing Phase II NVP metabolites and upregulating Phase II enzymes and *MRP7*. Accordingly, NVP-exposure led to decreased glutathione availability and alterations in the intracellular dynamics disfavoring free reduced glutathione and glutathionylated protein pools. Overall, these results demonstrate the adequacy of the 3D-HLC model for studying the bioactivation/metabolism of NVP representing a further step to unveil toxicity mechanisms associated with glutathione net flux changes.

## 1. Introduction

The well-known drawbacks of primary hepatocytes and hepatic cell lines regarding long-term metabolic stability and competence has prompted efforts for the development of more reliable in vitro hepatotoxicity models [1]. From those efforts, the differentiation of human iPSCs (induced pluripotent stem cells), ESC (embryonic stem cells) or hMSC (mesenchymal stem cells) into hepatocyte-like cells (HLC) have shown promising results. While various systems have been established to study liver pathology and toxicity, these models still lack some key liver-specific functionalities, and none of them has yet been validated for routine hepatotoxicity testing [2] which is of paramount importance to address drug-induced liver injury (DILI). Importantly, for proper benchmarking of in vitro liver-models it is crucial the definition of a set of consensus criteria, comprising critical elements such as cell viability, morphology, functionality and toxicological characterization, as proposed by Vinken and Hengstler [3].

Our team has previously established an improved hepatic differentiation protocol for deriving HLCs from human neonatal MSCs (hnMSCs) under 2D culture conditions [4]. The hepatic phenotype of the HLCs was confirmed by an unbiased whole genome analysis that placed the HLCs between human primary hepatocytes (hpHep) and HepG2, and distant from hnMSCs. Notably, the resort to spheroid or to multicompartment membrane bioreactor cultures further improved HLC maturation and functionality [5]. Indeed, three-dimensional (3D) culture systems allow higher cell-to-cell and cell-to-matrix contact, as well as nutrient and oxygen gradients and cell polarization [6,7,8,9]. These features are essential for the functional specialization of hepatocytes, which can be of enormous impact when employing in vitro systems to investigate susceptibility to drug-induced adverse-reactions [1], such as those induced by the hepatotoxic drug nevirapine (NVP), a non-nucleoside reverse transcriptase inhibitor [6]. In this sense, 3D stem cell-derived hepatic models may be promising tools for the early assessment of new molecules efficacy or detection of their potential toxic effects, with higher accuracy and decreased number of sacrificed animals [5,9,10]. 

Glutathione, a γ-glutamyl-L-cysteinyl-glycine tripeptide, is the most abundant intracellular antioxidant system, with functions such as cysteine reserve, participation in redox signaling and, most importantly, in metabolism and detoxification of xenobiotics and endogenous compounds [11]. Glutathione facilitates the plasma membrane transport of metabolites by different mechanisms, the most important of which is the formation of glutathione S-conjugates. However, glutathione conjugation reduces its availability and may compromise its normal functions. For glutathione production, the enzyme glutamate-cysteine ligase (GCL) binds glutamate and cysteine in a rate-limiting step of the synthetic pathway. This dipeptide is then covalently linked to glycine generating glutathione. Glutathione can be present in the reduced (GSH) and oxidized (GSSG) free forms or generate S-glutathionylated proteins upon post-translational modification (GSSP). The GSH/GSSG redox couple has the essential function of allowing reversible glutathionylation of proteins [12] and GSSP also prevents irreversible protein inactivation under oxidative stress insults, playing an important role regarding adaptive responses to drug-induced hepatotoxicity [13].

As mentioned above, NVP is an antiretroviral drug associated with severe hepatotoxicity, through the formation of reactive metabolites (Figure 1) [7]. NVP biotransformation comprises the formation of multiple Phase I metabolites (2-, 3-, 8-, and 12-OH-NVP, and 4-carboxy-NVP) that undergo subsequent Phase II glucuronidation or undergo detoxification trough GSH conjugation (Figure 1) [14]. NVP bioactivation pathways have also been proposed, namely the SULT1A1-mediated Phase II sulfonation of 12-OH-NVP to 12-sulfoxy-NVP; and the formation of NVP quinone methide from NVP, 12-OH-NVP and 12-sulfoxy-NVP. The reactive electrophilic metabolite NVP quinone methide has been proposed to be involved in the onset of NVP-induced liver toxicity through formation of covalent adducts with key proteins (Figure 1) [15]. Besides its extensive metabolism through Phase I and II enzymes, NVP has also the capability for induction of its own metabolism [14]. Indeed, NVP induces CYP2B6 and CYP3A4 [16], whereas its Phase I metabolite, 2-OH-NVP (Figure 1), induces SULT1A1 [6]. Therefore, due to its bioactivation, and subsequent detoxification and toxicity mechanisms, NVP has been shown to be a relevant drug for investigating drug metabolism and bioactivation competence of in vitro models [6,7].

In this context, we aimed a step forward and propose a novel integrated assessment of NVP metabolism coupled with the *glutathiolomic* profile to unveil the suitability of a human HLC 3D model as an alternative in vitro tool for human metabolism studies, and therefore appropriate for investigating mechanisms of DILI. To evaluate the competence of this model, we analysed (i) the formation of NVP metabolites, its effect on (ii) gene expression of CYPs and (iii) activity of Phase I and Phase II enzymes and hepatic transporters and its (vi) *glutathiolomic* profile, over long-term cultures. 

## 2. Results

### 2.1. NVP Modulates Key Biotransformation Enzyme Activity in 2D and 3D HLC Cultures

The formation of the major NVP metabolite, 12-OH-NVP, is credited mostly to CYP3A4, but also to CYP2D6 and CYP2C9 (Figure 1) [16]. On the other hand, the formation of 2-OH-NVP and 3-OH-NVP is exclusively mediated by CYP3A and CYP2B6, respectively, whereas the conversion to 8-OH-NVP is attributed to a group of subfamilies that include CYP3A4, CYP2B6 and CYP2D6 (Figure 1) [16]. Therefore, to evaluate the biotransformation competence of the HLCs, the expression level of genes relevant for NVP metabolism, as well as Phase I and II activities and corresponding NVP metabolites were assessed in the cultures treated with NVP for 3 (D27) and 10 days (D34) (Figure 2). The NVP concentration of 300 µM was used in this study after confirming the absence of cytotoxic effects in HLCs in all tested endpoints (data not shown). 

As shown in Table 1 and in Figure 3, Phase I and II activities were higher in 3D-HLCs when compared to 2D-HLCs, except for ECOD activity (7-ethoxycoumarin *O*-deethylation, covering CYP2B6, 1A1/2 and 2E1 [17]) at D27. In particular, CYP3A4/5 activity was induced 5-fold and 1.5-fold in 3D-HLC cultures at D27 and D34, respectively; whereas 2D-HLC cultures showed a ~1.5-fold induction at both time points (Figure 3). In 3D-HLC cultures, ECOD activity also showed a 1.3-fold and 3.7-fold induction after 3 (D27) and 10 days (D34) of treatment with NVP, respectively. Conversely, monolayer cultures only displayed ECOD induction (2.5-fold) at 3 days after treatment (Figure 3). *CYP2D6* gene expression level was slightly increased upon NVP treatment (Figure 2A). In line with these findings, the levels of NVP Phase I metabolites, together with the corresponding Phase I gene expression, were also elevated in 3D-HLCs (Figure 4). The conversion of NVP into 2-OH-NVP (total fraction) was superior in 3D-HLCs (Figure 4). Moreover, the formation of 12-OH-NVP (total fraction) was enhanced in both cultures from D27 to D34 (Figure 4). The amount of 3-OH-NVP duplicated from D27 to D34 in 3D-HLCs, whereas, as expected, the NVP minor metabolite, 8-OH-NVP, was barely detected in our systems (Figure 4A). 

Regarding Phase II metabolism and transporters expression levels, the 3D-HLCs cultures showed an improved performance as well. *UGT1A1*, *SULT1A1*, *GSTA1-A2* and *MRP7* overexpression was only observed in 3D-HLCs (Figure 2B–D). UGT activity and the glucuronic acid conjugates increased from D27 to D34 in 3D-HLCs (Figure 3), whereas no conjugates were detectable in 2D-HLCs at D34 (Figure 4A). SULT activity improved in both culture systems over time (Figure 3), resulting in increased production of 3 and 8-OH-NVP sulfate conjugates (Figure 4A).

Overall, NVP induced the expression and the activity of the enzymes responsible for its own metabolism, particularly in 3D culture conditions (Table 1). The longer treatment period allowed us to cover the induction period of NVP. This is confirmed by the relative proportions of metabolites in both systems (2-OH-NVP > 12-OH-NVP > 3-OH-NVP > 8-OH-NVP) and its trend to decrease the levels of 2-OH-NVP and increase the levels of 12-OH-NVP with the time of culture (Figure 4B). Moreover, the same trend has been observed in an additional timepoint (D31, data not shown).

### 2.2. HLC 3D Cultures Are More Efficient in Maintaining the Dynamics of Glutathione Pools 

Considering the role of GSH in detoxification of xenobiotics and/or their metabolites, we aimed to study the HLCs’ glutathione metabolism at different time points to further evaluate cells’ capacity to adjust to drug exposure. Herein, the *glutathiolomic* profiling in non-treated cells was firstly performed in order to compare basal glutathione net flux in 2D and 3D models. 

Total intracellular availability was quantified as the sum of contributions of the pools (free reduced (GSH) plus free oxidized (GSSG) and protein bound (GSSP)). This strategy allowed us to disclose basal intracellular availability of glutathione and its precursors (Figure 5A). Likewise, we clarified the intracellular dynamics of glutathione pools between free form (GSH + GSSG) and in the form of S-glutathionylated proteins (GSSP) (Figure 5B). 

In 2D-HLCs model, culture time (from D27 to D34) resulted in an increase in the intracellular cysteine levels and decreased GSSP (Figure 5). This may indicate that 2D cells might be striking to maintain intracellular reduced glutathione levels. The increase in cysteine level in 2D-HLCs was not followed by an increase in glutathione availability (Figure 5A), neither by increased formation of its precursor GluCys (Figure 5A). Therefore, the slight increase of intracellular free glutathione availability at D34 might not only be at the expense of decreasing GSSP (Figure 5B) to compensate for the incapacity of maintained glutathione synthesis over time (Figure 5A), but rather due to a decrease in gluthathione excretion at this timepoint (Figure 5A).

Finally, we determined the ratios of glutathione synthesis and extracellular catabolism (see CTRLs, Figure 6A). Culture time resulted in a shift from glutathione production to catabolism from D27 to D34 in 2D-HLCs (Figure 5A). In 3D-HLCs, on the other hand, the equilibrium GSH synthesis/catabolism is maintained more stable independently of the time in culture (there is no inter-day variability). The intracellular availability of GSH precursors and the distribution of GSH between its free and protein bound forms were maintained, which supports 3D-HLCs as a more relevant model to study the drug-induced *glutathiolomic* profile. 

Overall, data indicates that 3D cells were more efficient than 2D model in maintaining the dynamics of glutathione pools and therefore its *glutathiolomic* profile independently of cell culture time. 

### 2.3. The Glutathiolomic Profile of 3D Cultures Is Altered upon NVP Exposure in a Time-Dependent Manner

The effect of NVP exposure on the *glutathiolomic* profile of cells showed no changes on the synthesis nor catabolism of glutathione in the 3D model (Figure 6A). 2D had a different glutathione dynamic with higher catabolism. However, 3D presented a high reduction of glutathione in both free and protein bound forms at D34 (Figure 6C). NVP promoted a decrease of GSH intracellular precursors and increase in its main catabolism metabolite CysGly (Figure 6B), which is consistent with a decrease in both free and protein-bound intracellular fractions of GSH (Figure 6D), that was not observed in 2D model where no variations were observed.

Summing up, our findings showed that *glutathiolomic* profile of 3D cultures is altered upon NVP exposure in a time-dependent manner characterized by a reduction in net intracellular glutathione availability, a higher excretion of glutathione and a change in glutathione dynamics disfavoring its GSSP pools.

## 3. Discussion

The search for physiologically relevant in vitro hepatic models of human origin that reproduce liver complexity and that allow for long-term studies remains an important topic. Bearing in mind the need to overcome the interspecies differences (between rat and human) and the low availability of human hepatocytes, the objective of the present work was to evaluate the usefulness of a human-based 3D in vitro stem cell derived-hepatic model for long-term drug metabolism and bioactivation studies. Indeed, 2D cultures do not recapitulate the 3D arrangement nor the vascularization typical of the liver physiological environment [18]. 3D cultures of human or rat primary hepatocytes, as well as of hepatic cell lines and HLCs have shown their superiority relative to 2D cultures at the functional level, also allowing long-term cultures [5,6,19,20,21,22,23]. In particular, the organ-on-a-chip technology emerged as promising in vitro models by offering the possibility of mimicking blood circulation by creating a laminar fluid flow, seeding high cell densities, mimicking tissue architecture in a three-dimensional fashion, consuming low cell and reactant amounts and allowing interconnected culture of different cell types [24,25]. However, until today, few authors performed a thorough study on drug metabolism in such devices [26,27]. One of the biggest challenges of using organ-on-a-chip technology is still the high adsorption of small molecules to the microfluidic device materials (e.g., PDMS) requiring additional studies to allow the comparison with already existing data generated using polystyrene as material [26,28]. Spheroid based cultures, on the other hand, are quite well characterized and have the advantage of enabling both miniaturization and higher throughput [1,9,10,19].

Several 3D in vitro liver models have been reported in the literature over the past years [5,21,23,29,30,31,32,33,34]. However, a comprehensive and systematic comparison between distinct cell culture systems that would allow its wide adoption for pharmacological and toxicological applications is scarce, whereas data from stem cell derived HLCs is even scarcer. Importantly, most hepatic differentiation protocols do not generate fully mature hepatocytes with respect to a diversity of mature hepatic functions, including drug metabolizing capacity, albumin production, urea cycle activity, or glycogen storage ability. In addition, the time in culture is frequently overlooked, despite the reports supporting that cells need time in culture to mature and re-establish key ultrastructural and physiological traits [35]. In fact, the comparison of cell culture systems and cell sources is a challenging task. Cell viability and stability, morphological and architectural features, Phase I and Phase II metabolic capacity, response to a large panel of well-accepted reference drugs, as well as the physiological preservation of key metabolic and signaling pathways under long-term hepatic cultures shall be taken into consideration for pharmacological and toxicological studies. Altogether, these topics highlight the need for the proper definition of a set of consensus criteria, comprising critical elements such as cell viability, morphology, functionality and toxicological characterization, as well as time in culture [3]. It further supports that the choice of reference drugs for showing the relevance of the cell culture model and of the cell source is neither trivial nor consensual.

NVP was considered as model drug due to its toxicity mechanisms and given that its bioactivation requires a functional and physiologically relevant cellular model (showing Phase I, II and III activity). This drug can be bioactivated into the reactive electrophilic metabolites 12-sulfoxy-NVP (stemming from sulfotransferase-mediated sulfonation of 12-OH-NVP) and NVP-quinone-methide (formed upon NVP oxidation) (Figure 1). These metabolites have distinct abilities to form covalent adducts with proteins, which can either lead to toxic effects [36,37] or undergo detoxification via glutathione conjugation [6]. Thus, NVP reactive metabolites can also directly fine-tune glutathione availability and dynamics [7].

The steady state relative proportions of NVP metabolites in clinical, pre-clinical and in vitro studies differs greatly in the literature (Table 2). Nevertheless, 12-OH-NVP is typically described as the major NVP metabolite. 3-OH-NVP is consensually the third most detected metabolite, whereas the 8-OH-NVP is a minor metabolite [14,16] only detected in samples derived from human systems (Table 2), which is consistent with the low levels of 8-OH-NVP found in this study. In this work, as well, the most detected metabolite was 2-OH-NVP (~60% at D34), followed by 12-OH-NVP (~27% at D34), 3-OH-NVP (10–15% at D34) and 8-OH-NVP with a proportion of less than 2% in all conditions. Concerning the relative proportions of 12-OH-NVP and 2-OH-NVP, and although in our system the most abundant metabolite was 2-OH-NVP, there was a 3-fold increase of 12-OH-NVP from D27 to D34 (Figure 4). This result is in accordance with Erickson et al. [16] that showed that the formation rate of 2-OH-NVP by human CYP3A4/5 in microsomal preparations is ~2-fold higher than that of 12-OH-NVP. The decrease of 2-OH-NVP (2-fold decrease) and increase of 12-NVP-OH (3-fold increase) C_max_ from single dose to steady-state has also been reported in clinical studies [38,39], suggesting that our data might correspond to the NVP induction period that lasts at least two weeks in vivo [40]. Importantly, the 2-OH-NVP levels in the 3D-HLCs became comparable to those in the 3D-rat hepatocyte system after 3 days of NVP exposure. 

Regarding Phase II metabolism, the 3D conditions potentiated Phase II enzymes expression (*GSTA1-A2*, *UGT1A1* and *SULT1A*1) and activity (SULT and UGT) resulting in higher amounts of NVP-sulfate and glucuronic acid conjugates. Moreover, due to the relevance of S-conjugation to the toxicity/detoxification balance, *GSTA1-A2* expression levels were quantified and analysed in the context HLCs’ redox homeostasis and glutathione biosynthesis/degradation, by assessing HLC’s *glutathiolomic* profile upon NVP treatment. GSTs (Glutathione-S-transferases) are of major importance in the liver as a cellular defense mechanism and are responsible for the enzymatic conjugation with GSH. GSTA1/2 share both the expression and regulation mechanisms and substrates, and account for 3% of total hepatocyte cytoplasmic proteins [43]. 12-Sulfoxy-NVP and the arene oxide precursor of 3-OH-NVP are the only metabolites known to form glutathione conjugates [44]. From those, only the formation of 12-sulfoxy-NVP-derived glutathione adducts have been reported to be GST-dependent, mediated by GSTA1-1, GSTM1-1 and GSTA3-1 [44]. In this study, NVP induced *GSTA1-A2* in 3D-HLCs but not in 2D-HLCs, which may be a cellular adaptive mechanism for protection against oxidative stress [39]. Indeed, cytoprotective agents induce GSTA2 and concomitantly activate the PI3K-Akt/ERK-RSK1-mTOR pathways that in turn activate transcription factors favoring cell viability [39]. Accordingly, NVP altered our 3D model *glutathiolomic* profile in a time-dependent way. The NVP exposure time led to a reduction in net intracellular glutathione availability, a higher excretion of glutathione, probably in order to supply cysteine to maintain cell viability, and by a change in glutathione dynamics disfavoring GSSP pools, corroborating a toxic response. Studies are emerging showing that protein glutathionylation underlies the mechanism and might represent a predictive marker of drug-induced hepatotoxicity [13,45,46]. These major changes in glutathione net flux were observed only in 3D-HLCs, corroborating the competence of the cellular model to bioactivate NVP into its toxic metabolites. This observation and the maintenance of a stable basal *glutathiolomic* profile in the 3D model, supports the suitability of the 3D HLC model to investigate changes in net flux of glutathione, its availability and dynamics between intracellular pools upon toxic chronic insult. Moreover, it highlights the relevance of evaluating the global *glutathiolomic* profile comparing to GSH/GSSG ratio.

The transport of xenobiotics and its metabolites is as relevant as Phase I and II metabolisms for assessing drug toxicity profiles. This aspect is often overlooked on HLCs studies. NVP is a MRP-7 (ABCC10) substrate [47] and although there is no report for *MRP7* induction by NVP in human liver, its induction has been observed in in vivo mouse models of cholestasis (lipopolysaccharide administration and bile duct ligation) revealing its importance on hepatotoxicity [48]. A recent in vitro study using an organotypic culture of cryopreserved hpHep exposed to supra-therapeutic concentrations NVP also revealed an increased cholestasis potential of NVP through a transcriptomic analysis [49]. Our previous transcriptomic study on 2D-HLCs has shown that *MRP7* levels in at D34 were closer to hpHep than the expression levels observed in HepG2 and in hnMSCs [4]. In the present study, we observe that *MRP7* gene expression was induced by NVP, but only in 3D-HLCs. Taken together, our 3D-HLCs were able to show a complete metabolic profile covering all essential process of drug metabolism.

## 4. Materials and Methods

### 4.1. Cell Culture

All culture media and supplements, solvents (all of analytical grade) and other chemicals were acquired from Sigma-Aldrich (Madrid, Spain) unless specified. 

hnMSCs isolated from human umbilical cord stroma were fully characterized and expanded in Eagle’s minimum essential medium - alpha modification (α-MEM) supplemented with 10% (*v*/*v*) of fetal bovine serum (FBS, Gibco, Paisley, UK) as described previously [50]. For generating HLCs, a three-step differentiation protocol was applied to hnMSCs cultured in 2D and as 3D spheroids using ultra-low attachment (ULA, VWR International, Radnor, PA, USA) plates as detailed previously [5]. Briefly, hnMSCs were seeded at a density of 1.5 × 10^5^ cells/cm^2^ in a rat-tail collagen (0.2 mg/mL)-coated surface. At day 17 (D17) of differentiation, cells were trypsinized and reinoculated in differentiation medium (Iscove’s modified Dulbecco’s medium (IMDM) with 8 ng/mL oncostatin M (OSM; Peprotech, Rocky Hill, NJ, USA), 1 µM dexamethasone, 1% DMSO and 1% insulin-transferrin-selenium solution (Gibco, Grand Island, NY, USA) at a final concentration of 1.72 µM, 68.75 nM and 38.73 nM, respectively) containing 20 µM 5-azacytidine and 5% (v/v) FBS into (i) ULA plates (2.5 × 10^4^ cells/mL) to obtain a 3D spheroid culture or (ii) 2D culture plates pre-coated with collagen (2 × 10^4^ cells/cm^2^) as control (Figure 7). Cells were maintained for 24 h and then the medium was replaced by third step differentiation medium [5]. Medium replacement occurred every 3 days. The cell cultures were monitored microscopically using phase contrast microscopy (Olympus CK30 inverted microscope, Tokyo, Japan).

For NVP (Cipla, Mumbai, India) treatment, a stock solution was prepared in DMSO at a concentration of 150 mM. For cell incubation, NVP was diluted in complete medium to a final concentration of 300 µM. This concentration is considered non-cytotoxic and supra-therapeutic and was chosen according to values previously described in the literature [6,8,51,52,53]. HLCs at D24 of the differentiation protocol were exposed during 3 (D27) and 10 days (D34), whereas hnMSCs were treated for 72 h (Figure 7). Cell culture volume was adjusted in order to obtain a volume per cell equivalent in both 3D and 2D systems. Complete medium without NVP, as well as complete medium with solvent (supplemented with additional 0.2% DMSO) and without NVP were used as negative and solvent controls, respectively.

Images were acquired using Moticam 5.0 (Motic, Barcelona, Spain) and recorded with Motic Images Plus 2.0 software. The spheroid diameter was evaluated every 3 days [50].

The cell number was calculated as described previously [19,21]. The protein content of the cell pellet samples from 3D and 2D cultures was determined using the BCA protein assay kit (Millipore, Burlington, MA, USA) according to manufacturer’s instructions. 

### 4.2. Nevirapine Cytotoxicity

NVP cytotoxicity was assessed using the MTS cell viability assay (Promega, Madison, WI, USA), according to the manufacturer’s instructions. NVP concentrations ranging from 25 to 2000 µM were tested in HLCs. Briefly, HLCs were seeded in 100 μL of culture medium at a density of 6400 per well in 96-well collagen-coated plates. Cultures were then incubated for 72 h. Three independent experiments were performed. The results showed no differences between the NVP-exposed cells and non-exposed cells up to 1000 µM indicating no cytotoxic effects of either NVP or its metabolites during the whole culture time for the concentration of 300 µM.

### 4.3. Biotransformation Activity

CYP enzyme activity was measured by means of 7-ethoxycoumarin-*O*-deethylase (ECOD) activity (covering CYP2B6, 1A1/2 and 2E1) [17] and P450-Glo^TM^ assays (Promega) and UGT activity by means of 4-methylumbelliferone conjugation as described previously [4]. The thermostable SULT1A1 activity was assessed in cell homogenates from both 2D and 3D cultures, obtained by sonication of cell pellets, containing ~7 × 10^4^ cells, suspended in 200 µL of sonication buffer (10 mM triethanolamine at pH 7.4, with 250 mM sucrose and 5 mM β-mercaptoethanol). Sonication procedures were performed on ice. SULT1A1 activity was assessed using the 2-naphthol (2-NAF) sulfonation assay in cell homogenates as previously described [6]. The reaction mixture contained 50 mM phosphate buffer (pH 7.5), 5 mM MgCl_2_, 500 μM EDTA, 20 µM 3′-phosphoadenosine-5′-phosphosulfate (PAPS), 5 mM potassium *p*-nitrophenyl sulfate (KNPS) and 100 µM of 2-naphthol (Merck, Darmstad, Germany). An effective PAPS-regenerating system was obtained with the addition of *p*-nitrophenyl sulfate. SULT1A1 catalyzes the synthesis of 2-naphthyl sulfate from 2-NAF and PAPS, while regeneration of the PAPS cofactor produces *p*-nitrophenol. The measurement of *p*-nitrophenol absorbance at 405 nm is, therefore, an indirect measurement of sulfotransferase activity. A calibration curve was generated with *p*-nitrophenoland the absorbance was registered at 405 nm (SPECTROstar Omega, BMG Labtech, Ortenberg, Germany) after a 5-h incubation at 37 °C in a humidified atmosphere.

The ECOD, CYP3A4/5, UGT and SULT1A1 activities were normalized to incubation time (h) and cell number (10^6^ cells) and expressed as fold induction in NVP-treated cultures relative to non-treated cultures. The data for the activity normalized to incubation time (h) and cell number (10^6^ cells) is presented in Appendix A.

### 4.4. Quantification of Nevirapine Metabolites 

Supernatants from 3D and 2D cell cultures were collected (triplicates) on D27, D31 and 34 of the differentiation and treated separately. The data for D31 is presented in Appendix A. Phase I and Phase II NVP metabolites were analyzed and quantified by HLPC as described previously [41]. Briefly, for the quantification of Phase I and Phase II NVP metabolites, 15 mL of cell culture supernatants were incubated for 24 h at 37 °C in the absence (unconjugated/free metabolites) or presence of either sulfatase (type H-1 from *Helix pomatia*, E.C. 3.1.6.1, 100 U/mL,) and d-saccharic acid 1,4-lactone (4 mg/mL), for the detection of free metabolites plus sulfate conjugates [54], or β-glucuronidase (type VII-A from *E. coli*, E.C. 3.2.1.31, 100 U/mL), for the detection of free metabolites plus glucuronic acid conjugates [55]. The pH of all samples was readjusted to 7.0 before extraction of the free metabolites with dichloromethane (2 × 15 mL). Upon evaporation, the organic extracts were dissolved in 150 µL H_2_O/methanol (1:1) and analyzed on an Agilent 1100 Series HPLC-UV system (Agilent Technologies Inc., Santa Clara, CA, USA) as described in Marinho et al. [41]. The metabolite levels were normalized to the cell number and expressed as ng/10^6^ cells.

### 4.5. Gene Expression

RNA isolation was performed using Trizol^®^ (Ambion, Inc., Austin, TX, USA). RNA was quantified prior to cDNA synthesis as previously described. cDNA synthesis was performed from 0.5 µg of RNA using a commercially available kit (NZYTech, Lisbon, Portugal) and quantitative real-time PCR (qPCR) was performed using PowerUp™ SYBR^®^ Green Master Mix (Applied Biosystems^®^/ Life Technologies, Austin, TX, USA), according to the manufacturer’s instructions. A final reaction volume of 15 µL, with 1 µL of template cDNA and 0.1 µM of forward and reverse primers (Table 3) was used and the reaction was conducted in the ABI7300 Real-time PCR system (Applied Biosystems^®^/Life Technologies, Carlsbad, CA, USA). The amplification of cDNA consisted of a denaturation step at 95 °C for 10 min, 40 cycles of denaturation at 95 °C for 15 s, annealing at 60 °C for 1 min and extension at 72 °C for 30 s. As a quality and specificity measurement, a dissociation stage was added to determine the melting temperature in all runs. The comparative Ct method (2^−ΔΔCT^) was used to quantify the amount of target genes, normalized to the reference gene β*-ACTIN* and relative to non-NVP-treated cells, unless otherwise stated. The efficiency of each PCR reaction was estimated from a set of serially diluted cDNA control solutions from in order to construct a standard curve (1, 10^−1^, 10^−2^) for each gene, considering efficiencies between 1.8–2.2.

### 4.6. Glutathiolomic Profiling

Both cells (intracellular fraction) and cell supernatants (extracellular fraction) were collected from 2D and 3D cultures upon exposure or non-exposure of NVP. The extracellular (cell culture supernatants) and intracellular (cell lysates) fractions were collected at D27 and 34 of the differentiation protocols. The samples were processed and the *glutathiolomic* profile was measured as previously described [59], comprising the levels of cysteine (Cys), γ-glutamylcysteine (GluCys), cysteinylglycine (CysGly) and glutathione. The *glutathiolomic* profile comprises an integrated assessment of net availability and dynamics of glutathione in cell. In addition to the commonly used approach of evaluating reduced (GSH)/oxidized (GSSG) glutathione ratio, a complete assessment of glutathione net flux was performed. This included the clarification of total availability, distribution between intracellular pools, synthesis and degradation. For that, total (free + protein bound) intracellular glutathione availability was measured. Also, we addressed the dynamics of its intracellular distribution between pools: free (reduced and oxidized) and protein bound through the quantification of each fraction. Finally, we measured anabolic (intracellular GluCys/Cys ratio) and catabolic (extracellular CysGly/GSH ratio) activities for glutathione. Briefly, for *glutathiolomic* profiling cells were collected and immediately sonicated in 200 µL of ice-cold phosphate-buffered saline with 1% Triton. The amount of total thiols was obtained by reducing the sulfhydryl groups with tris(2-carboxyethyl)phosphine hydrochloride (TCEP; 100 g/L). After a 30 min incubation at room temperature, the samples were treated with trichloroacetic acid (TCA; 100 g/L) containing EDTA (1 mM), for protein precipitation. After centrifugation (13,000× *g*, 10 min, 4 °C), the supernatant was collected for derivatization of the free sulfhydryl groups. Derivatization was performed with 7-fluorobenz-2,1,3-oxadiazole-4-sulfonic acid ammonium salt (SBD-F) at 1 g/L in 125 mM sodium tetraborate buffer (pH 9.5) with 4 mM EDTA and 1.55 M NaOH, at 60 °C for 1 h, protected from light. Finally, a volume of 30 µL was analyzed by HLPC-FD. The quantification of the free forms was performed in cells and in cell culture supernatants. The latter were collected on ice and centrifuged for 5 min at 10,000× *g* at 4 °C for cell debris removal, whereas cell samples were prepared as abovementioned. Samples were then submitted to protein precipitation with TCA, with subsequent centrifugation (13,000× *g*, 10 min, at 4 °C) and reduced with TCEP for RLMWT quantification or incubated with reverse osmosis water to obtain the naturally reduced RSH fraction. After 30 min incubation at room temperature, the previously described derivatization protocol was followed. HPLC-FD analysis was performed on a Shimadzu LC-10AD VP system (Shimadzu Scientific Instruments Inc., Columbia, MD, USA) using a reversed-phase C18 LiChroCART 250-4 column (LiChrospher 100 RP-18, 5 µm, VWR International, Radnor, PA, USA), at 29 °C with a fluorescence detector (λ_Ex_: 385 nm/λ_Em_: 515 nm). The mobile phase consisted of 100 mM sodium acetate buffer (pH 4.5) and methanol [98:2 (v/v)]. The analytes were separated in isocratic elution mode for 22 min, at a flow rate of 0.6 mL/min. The data expressed as average ± SD were normalized with cell number and represented as nmol/10^6^ cells for non-treated cells and as fold induction in NVP-treated cultures relative to non-treated cultures.

### 4.7. Statistical Analysis

Data comparison was performed by two-way ANOVA analyses using GraphPad Prism version 6.0 (GraphPad Software, Inc., San Diego, CA, USA). Differences were considered statistically significant for *p* < 0.05.

## 5. Conclusions

The work herein presented shows for the first time that a 3D-HLC human model is able to metabolize NVP by correctly displaying its biotransformation profile, as well as by mimicking the auto-induction processes observed in patients treated with NVP-based regimens. Importantly, to our knowledge, this also the first report where a similar pattern regarding metabolites’ quantification is shown in human stem cell-derived HLCs. Moreover, an additional innovative aspect of this work, the *glutathiolomic* analysis, reflected the higher glutathione production in 3D cultures, which together with a more effective glucuronic acid conjugation also suggests a higher ability to scavenge reactive species in the 3D-HLCs compared to the 2D model. Altogether, these observations reinforced the 3D-HLCs biotransformation competence over the 2D model. In summary, this study constitutes a step forward in the development of human physiologically relevant mechanistic models for drug testing where an integrated approach might guide risk mitigation for early drug safety assessments during drug development.

## Figures and Tables

**Figure 1 ijms-21-03998-f001:**
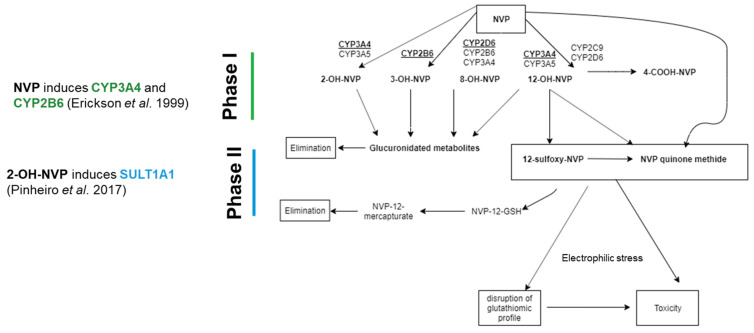
Schematic representation of Nevirapine (NVP) biotransformation and toxic metabolites formation.

**Figure 2 ijms-21-03998-f002:**
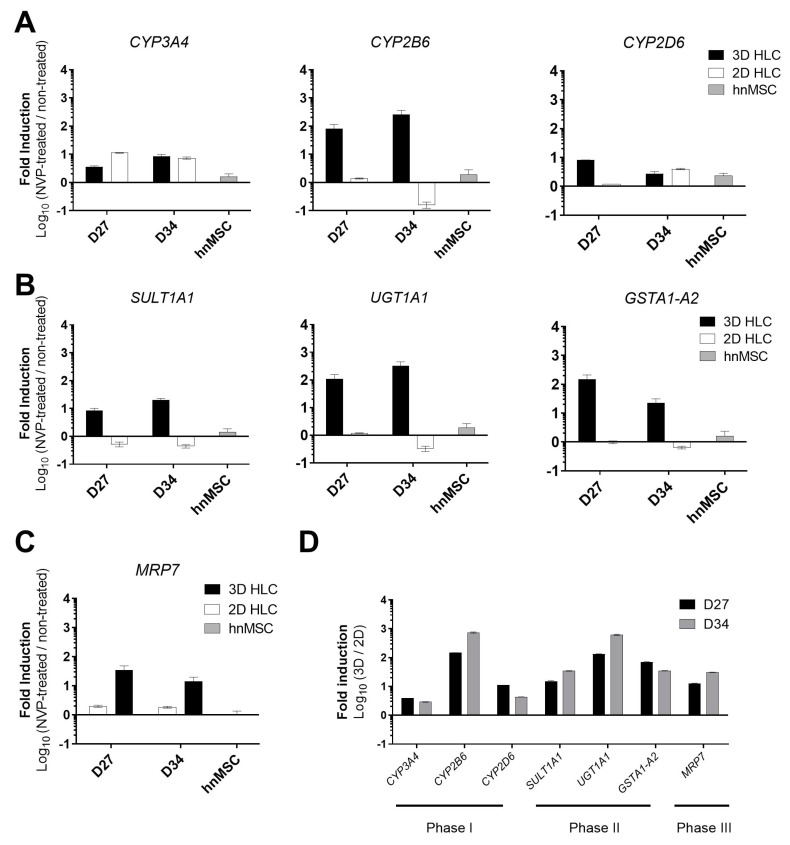
Gene expression analyses of Phase I, II and III related genes of 3D- and 2D-HLCs after 3 (D27) and 10 days (D34) of NVP treatment. (**A**) Gene expression of Phase I enzymes comprising *CYP3A4*, *CYP2B6* and *CYP2D6* in NVP-treated cells relative to non-treated cells. (**B**) Gene expression of Phase II enzymes comprising *SULT1A1*, *UGT1A1* and *GSTA1-A2* in NVP-treated cells relative to non-treated cells. (**C**) Gene expression of the *MRP7* transporter in NVP-treated cells relative to non-treated cells. (**A**–**C**) Data for hnMSC exposed to NVP for 3 days and used as negative controls, are also shown. (**D**) Phase I, II and III gene expression in 3D-HLCs relative to 2D-HLCs in NVP-treated cells at D27 and D34. The data (Mean ± SD, *n* = 3) are normalized to the reference gene *β-ACTIN* and expressed as the log_10_ of the ratios.

**Figure 3 ijms-21-03998-f003:**
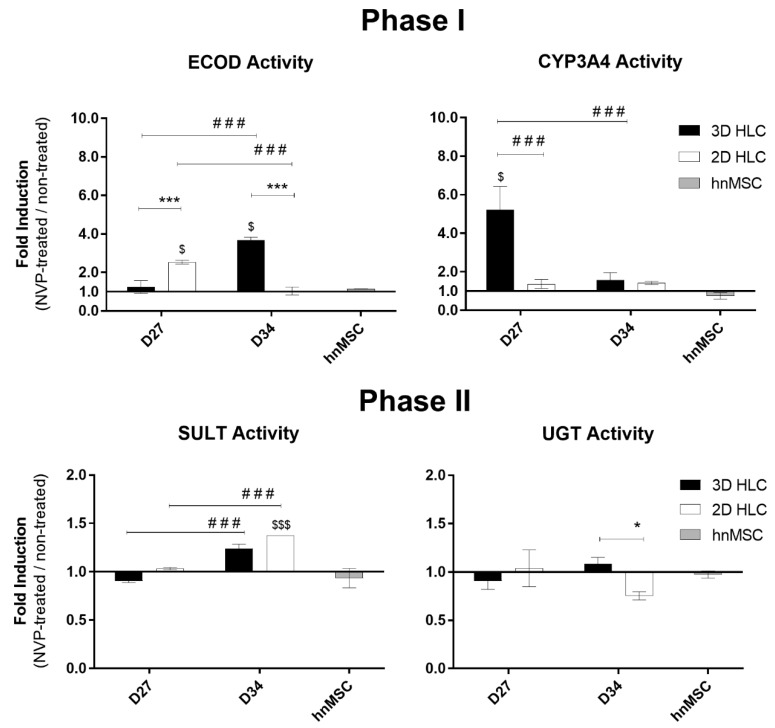
Phase I and Phase II enzymatic induction in 3D- and 2D-HLCs after 3 (D27) and 10 days (D34) of NVP treatment. The data for hnMSC, cultured in monolayer and exposed to NVP for 3 days, are presented as negative control. * *p* < 0.05 and *** *p* < 0.001 relative to 2D-HLCs, ^$^
*p* < 0.05 and ^$$$^
*p* < 0.001 relative to non-treated and ^###^
*p* < 0.001 relative to D27 of culture. The data (Mean ± SD, *n* = 3) are represented as fold induction in NVP-treated cells relative to non-treated cells.

**Figure 4 ijms-21-03998-f004:**
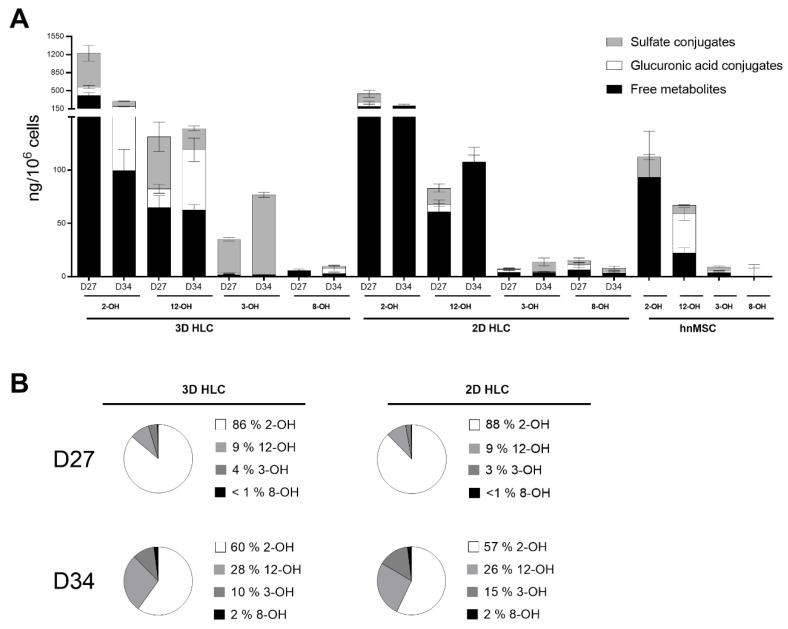
(**A**) Levels of NVP Phase I and II metabolites in 3D- and 2D-HLCs after 3 (D27) and 10 days (D34) of NVP treatment. hnMSC cultured in monolayer and exposed to NVP for 3 days are also shown and (**B**) relative proportions of total NVP metabolites (free metabolites + sulfate conjugates + glucuronic acid conjugates) at D27 and D34. The data (Mean ± SD, *n* = 3) are normalized to total cell number.

**Figure 5 ijms-21-03998-f005:**
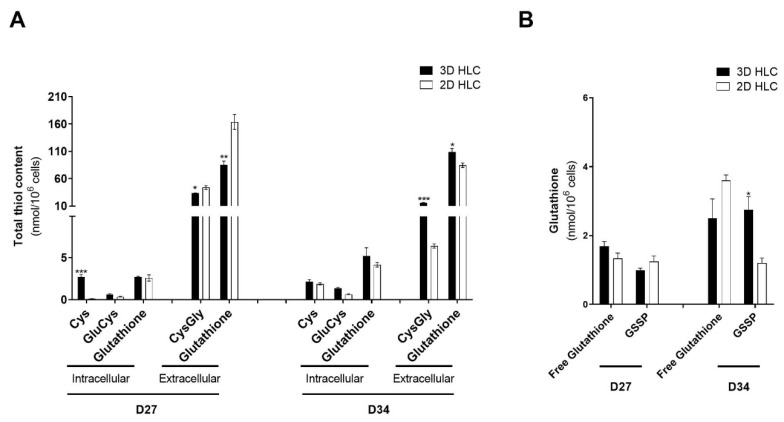
*Glutathiolomic* analysis of 3D- and 2D-HLCs at D27 and D34. (**A**) Total intracellular availability for cysteine (Cys), glutamylcysteine (GluCys) and glutathione and total extracellular levels of cysteinylglycine (CysGly) and glutathione; (**B**) Intracellular levels of protein-bound glutathione (GSSP) and free glutathione; * *p* < 0.05, ** *p* < 0.01 and *** *p* < 0.001 relative to 2D-HLCs. The data are represented as Mean ± SD, *n* = 3.

**Figure 6 ijms-21-03998-f006:**
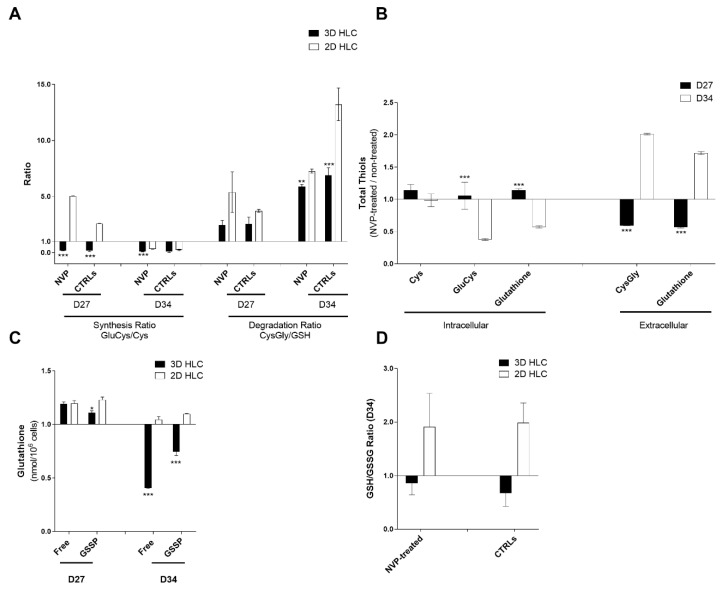
*Glutathiolomic* analysis of 3D- and 2D-HLCs treated or non-treated with NVP. (**A**) Effect of NVP on glutathione synthesis and degradation ratios, after 3 (D27) and 10 days (D34) of NVP treatment; (**B**) Effect of NVP on total intracellular availability for cysteine (Cys), glutamylcysteine (GluCys) and glutathione and total extracellular levels of cysteinylglicine (CysGly) and glutathione, in 3D-HLCs, after 3 (D27) and 10 days (D34) of NVP treatment; (**C**) NVP effect on intracellular levels of protein-bound glutathione (GSSP) and free glutathione, after 3 (D27) and 10 days (D34) of NVP treatment; (**D**) NVP effect on glutathione oxidation (intracellular GSH/GSSG ratio) after 10 days of NVP treatment (D34). * *p* < 0.05, ** *p* < 0.01 and *** *p* < 0.001 relative to 2D-HLCs. The data are represented as Mean ± SD, *n* = 3.

**Figure 7 ijms-21-03998-f007:**
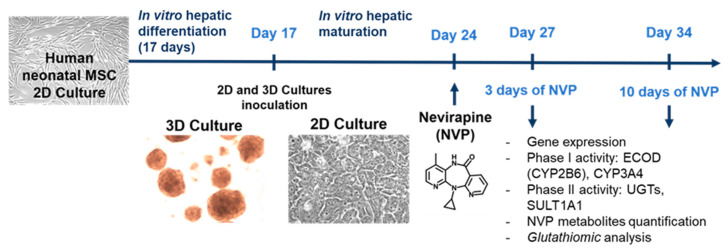
Schematic representation of the study design.

**Table 1 ijms-21-03998-t001:** Comparative summary of 3D- and 2D-HLC metabolic capacity upon 10 days of NVP treatment (D34).

Endpoint ^a^			3D-HLC (NVP Treated/ Non-Treated)	2D-HLC (NVP Treated/ Non-Treated)	3D-HLC/ 2D-HLC (NVP Treated)
Gene expression	Phase I	*CYP3A4*	+	+	+
		*CYP2B6*	+++	unchanged	+++
		*CYP2D6*	+	+	+
	Phase II	*SULT1A1*	+	unchanged	++
		*UGT1A1*	+	unchanged	+++
		*GSTA1-A2*	+	unchanged	++
	Phase III	*MRP7*	+	unchanged	++
Enzymatic activity	Phase I	ECOD	++	unchanged	+
		CYP3A4	unchanged	unchanged	unchanged
	Phase II	UGTs	unchanged	unchanged	+
		SULT1A1	unchanged	unchanged	unchanged
NVP metabolites ^b^	Phase I	2-OH-NVP	detected	detected	+
		(CYP3A4)			
		12-OH-NVP	detected	detected	+
		(CYP3A4/2D6/2C9)			
		3-OH-NVP	detected	detected	++
		*(CYP2B6)*			
		8-OH-NVP	detected	detected	unchanged
		*(CYP3A4/2D6/2C9)*			
	Phase II	Sulfate conjugates	detected	detected	++
		Glucuronic acid conjugates	detected	not detected	+++
*Glutathiolomic* profile		GSH synthesis	residual	residual	+
		GSH catabolism	+	++	−

^a^ Gene expression: (+) 1–20-fold increase, (++) 20–250-fold increase, (+++) >250-fold increase; Enzymatic activity, NVP metabolites and glutathione content: (−) <1-fold decrease; (+) 1–4-fold increase; (++) 4–10-fold increase; (+++) >10-fold increase. ^b^ Detected/not detected: production of nevirapine metabolites in nevirapine-treated cultures.

**Table 2 ijms-21-03998-t002:** Comparison of relative levels (expressed as percentage) of NVP metabolites in clinical, pre-clinical and in vitro studies.

Study Type	Sample	2-OH-NVP *	12-OH-NVP *	3-OH-NVP *	8-OH-NVP *	Ref.
Clinical, 2 weeks of NVP treatment	Human plasma	13	43	39	2.0	[40]
Clinical, 4 weeks of NVP treatment	Human plasma	15	43	42	2.0	[40]
Clinical, Steady state NVP treatment	Human urine	23	33	33	1.0	[14]
Clinical, Steady state NVP treatment	Human plasma	13	80	6.0	0.0	[41]
Clinical, Steady state NVP treatment	Human plasma	16	75	7.0	2.0	[42]
Clinical, Single Dose NVP treatment	Human plasma	32	52	16	0.0	[38]
Clinical, Steady state NVP treatment	Human plasma	5	51	9	35	[38]
In vitro, spheroids of freshly isolated rat primary hepatocytes, exposed to NVP for 11 days	Cell culture supernatant of rat hepatocytes	25	57	18	0.0	[6]
In vitro, human cryopreserved microsomal preparations	Microsomes, reaction buffer	39	27	27	7.0	[16]
In vitro, HLC spheroids exposed to NVP for 10 days	Human cell culture supernatant	60	28	10	2.0	Present study

* the relative amounts of NVP metabolites were collected from the literature considering total amounts of metabolites, including Phase II metabolites that were incorporated as their Phase I form.

**Table 3 ijms-21-03998-t003:** Primers used for qRT-PCR characterization of NVP treated and non-treated HLCs, hnMSCs.

	Primers	Reference
*ACTB*_F	CATGTACGTTGCTATCCAGGC	PrimerBank ID 4501885a1 [56]
*ACTB*_R	CTCCTTAATGTCACGCACGAT	
*CYP3A4*_F	ATTCAGCAAGAAGAACAAGGACA	[57]
*CYP3A4*_R	TGGTGTTCTCAGGCACAGAT	
*CYP2B6*_F	TTCCTACTGCTTCCGTCTATCAAA	[58]
*CYP2B6*_R	GTGCAGAATCCCACAGCTCA	
*CYP2D6*_F	TGGCAAGGTCCTACGCTTC	[58]
*CYP2D6*_R	GCCACCACTATGCACAGGTT	
*SULT1A1*_F	CGGCACTACCTGGGTAAGC	PrimerBank ID: 29540539a1 [56]
*SULT1A1*_R	CACCCGCATGAAGATGGGAG	
*UGT1A1*_F	TGACGCCTCGTTGTACATCAG	[58]
*UGT1A1*_R	CCTCCCTTTGGAATGGCAC	
*GSTA1-A2*_F	TGCAACAATTAAGTGCTTTACCTAAGTG	[58]
*GSTA1-A2*_R	TTAACTAAGTGGGTGAATAGGAGTTGTATT	
*MRP7*_F	TGGCACATTCCCCTCATGG	[58]
*MRP7*_R	CCACAACACGGTCAGCACTA	

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
