# Peer review of "Nevirapine Biotransformation Insights: An Integrated In Vitro Approach Unveils the Biocompetence and Glutathiolomic Profile of a Human Hepatocyte-Like Cell 3D Model"

_ijms, 2020, doi:10.3390/ijms21113998_

Round 1
Reviewer 1 Report
In the current paper the authors aimed to study the usefulness of an in vitro 3D model of mesenchymal stem cell-derived hepatocyte-like cells (HLC) in the context of drug toxicity with the retroviral and hepatotoxic drug nevirapine (NVP). The authors used monolayer (2D-HLC) cultures as well as 3D spheroids (3D-HLC) and provide provide data on the formation of NVP metabolites, NVP´s effect on CYPs mRNA expression and enzyme activity, effects on hepatic transporters and the glutathione metabolism.
Overall the paper appears solid and sound. However, the presentation can be improved. In the graphs the line thickness is in general too low, which makes the graphs appear very faint.
Author Response
Reviewer 1
Comment 1: In the current paper the authors aimed to study the usefulness of an in vitro 3D model of mesenchymal stem cell-derived hepatocyte-like cells (HLC) in the context of drug toxicity with the retroviral and hepatotoxic drug nevirapine (NVP). The authors used monolayer (2D-HLC) cultures as well as 3D spheroids (3D-HLC) and provide provide data on the formation of NVP metabolites, NVP´s effect on CYPs mRNA expression and enzyme activity, effects on hepatic transporters and the glutathione metabolism.
Overall the paper appears solid and sound. However, the presentation can be improved. In the graphs the line thickness is in general too low, which makes the graphs appear very faint.
Response:
We thank the Reviewer for the suggestion. The line thickness of all graphs has been improved from ¼ to 1 to allow proper viewing of the graphs.
Reviewer 2:
Comment 2: Cipriano et al. present a well established model and reasonable results about in vitro hepatic metabolism of anti-HIV drug, nevirapine. Overall structure and results are well written and interesting. I have the following concerns for the authors to consider regarding this work.
Nevirapine is metabolized via CYP3A4/3A5, CYP2B6, CYP2D6, and CYP2C9 as authors mentioned. Nevirapine induces CYP2B6 and CYP3A4, but CYP2D6 is not inducible form in general condition. because CYP2C9 and CYP2D6 is quite important in general drug metabolism as major isoforms of CYP and in pharmacogenetics. Thus, the differences of 2-OH, 3-OH, 8-OH, 12-OH can be related with difference expression level of those CYP isoforms (as the referenced paper, ERICKSON et al. 1999). But, present research and former research paper from same research group did not show other CYPs activity. samples from two time points preparation is limited explain the consquence of enzyme induction including CYPs, SULTs, UGTs.
Response:
We agree with the Reviewer’s comment that NVP metabolism is credited mostly to CYP3A4. However, significant contributions from isoforms 2D6, 2C9 and 2B6 have been reported in the literature (Erickson et al., 1999; 10.1124/dmd.108.024851) as summarized in Figure 1. One important aspect is that NVP is also an auto-inducer, as referred before by Fan-Harvard et al., 2013 (10.1128/AAC.02294-12), and further demonstrated in the present work where the activity of both CYP3A4 and CYP2B6 is induced upon treatment with NVP (Figure 1).
In this work, we studied HLC’s metabolic profile upon NVP treatment. In this context, we evaluated the gene expression levels of CYP3A4, CYP2B6 and CYP2D6, as well as the CYP3A4 and ECOD (that reflects primarily the activity of CYP2B6 followed by CYP2E1 and CYP1A2) enzyme activities. Regarding Phase II metabolism both gene expression and enzyme activity of UGTs and SULTs is also shown, covering most of the enzymes involved in NVP metabolism. These assays were aimed at demonstrating the responsivity of the cells, demonstrating that metabolic enzyme expression and activity could be induced by NVP, resulting in the formation of the known NVP metabolites, similarly to what is observed with other in vitro models.
Importantly, as depicted in Figure 2D, the gene expression level of these important enzymes is higher in 3D-HLCs than in 2D-HLCs. In line with these findings, the levels of NVP Phase I metabolites, together with the corresponding Phase I and II enzyme activities, were also elevated in 3D-HLCs (Figures 4 and 3).
As such, although the data from D31 was also acquired, other timepoints of enzyme activity and expression were not initially included in the graphs. We chose to present only D27 and D34 for the sake of simplicity since it did not compromise any of the conclusions described. Nevertheless, the data from D31 is now included as supplementary data (Supplementary Figure 2) as referred in the methods section.
Comment 3: ECOD (courmain 7-hydroxylation) is more representative for CYP2A6 in human generally. (Pelkonen O, Rautio A, Raunio H, Pasanen M. CYP2A6: a human coumarin 7-hydroxylase. Toxicology. 2000 Apr 3;144(1-3):139-47.) authors mentioned ECOD represent CYP2B6 and CYP1A1/1A2, this is not appropriate substrate and metabolites for CYP2B6, 7-benzyloxy resorufin O-dealkylation or bupropion hydroxylation is better reaction for CYP2B6. Thus, induction of phase I enzyme (figure 3) is not enough to explain the metabolism of phase I metabolism of nevirapine. as same as gene expression level, CYP2D6 activity is required.
Response:
The assay and references that the Reviewer mentioned correspond to the coumarin 7-hydroxylation that is indeed specific for CYP2A6.
The methodology used in this study was the ECOD (7-ethoxycoumarin O-de-ethylase) activity assay, that is the O-de-ethylation of 7-ethoxycoumarin, resulting in the formation of 7-hydroxycoumarin, which is then readily estimated fluorometrically. We agree with the Reviewer that the ECOD assay reflects the activity of several CYPs. This has been clearly detailed in the literature. According to Wilkening et al. 2003 (10.1124/dmd.31.8.1035) "ECOD primarily reflects the activity of CYP2B6 followed by CYP2E1 and CYP1A2". This has also been reinforced by Yamazaki et al. 1996 (10.1016/0006-2952(95)02178-7), Wrighton et al. 1992 (10.3109/10408449209145319 ) and by Castell et al. 1996 (Castell, José V., and Maria Jose Gomez-Lechon, eds. In vitro methods in pharmaceutical research. Elsevier, 1996., pages 140-141), among others. This information has been detailed in the methods section (line 374). Nevertheless, in order to reinforce and clarify this information, this description has also been edited accordingly in the first time that the ECOD assay is mentioned in the manuscript, ie, in the Results Section, lines 119-122 and 123.
In the manuscript, we claim that there is induction of phase I (ECOD and CYP3A4 activities) and II enzymes activity, which is further supported by the formation and presence of the NVP specific metabolites. Moreover, the fact that NVP induces CYP3A4 and CYP2B6 is also in accordance with the results observed for the ECOD and CYP3A4 activities (Figure 3).
Comment 4: The concept of 'glutathiolomic' is very nice, but the consequence of glutathione catabolism in more dominants in kidney than liver via Gamma-glutamyltransferase (GGT). For the explain of glutathione metabolism, glutathione conjugate of nevirapine (eg.NVP-12-glutathione), cysteinyl/mercapturic acid conjugate metabolites levels and gamma-glutamylcysteine ligase, GGT mrp2 and mrp5 expression/activity are also required.
Response:
The authors acknowledge the Reviewer’s comment and would definitely consider this topic for future studies.
The focus of this work was to address glutathione dynamics in a more integrated approach, rather than to perform the classic measurement of the reduced/oxidized glutathione ratio. In this context, the metabolites and pools of glutathione metabolism, have been measured as an indirect of evaluating the presence and activity of the enzymes responsible for their formation.
Altough this was not the main aim of this work, the authors also agree that a more detailed study of downstream metabolite-glutathione conjugates would further support the formation and detoxification of the 12-sulfoxy-NVP. However, the induction of GSTA1-A2 at gene expression level also indicates a possible relevant role for glutathione conjugation beyond the glutathiolomic profile. The authors also agree that a thorough evaluation of MRP2 activity would support the glutathiolomic data since it is responsible for GSSG, glutathione S-conjugates and reduced glutathione. The presence of MRP2 in HLCs has already been confirmed previously by the authors at protein level in both 3D and 2D HLCs (Cipriano et al. 2017, Ref 5) and the gene expression of MRP5 and MRP2 were compared in a transcriptomic analysis in 2D HLCs with primary hepatocytes, where MRP5 (ABCC5) showed to be more expressed in HLCs than in the cell line HepG2 (Cipriano et al. 2017, Ref 4).
Finally, we would like to highlight that this study is, to the best of our knowledge, the first approach towards the glutathiolomic analysis (or glutathione metabolomics) in human HLCs and, therefore, the data presented here would support a variety of future studies.
Comment 5: It is better showing activities of CYPs in (eg. molar concentration per protein) in D27, D34 and control group, to show how comparable to standard procedure (primary hepatocytes)
Response:
We understand the Reviewer’s remark. However, it must be highlighted that the main aim of this manuscript was to evaluate the usefulness of a human based 3D in vitro HLC model for long-term drug metabolism and bioactivation in comparison with conventional 2D system, using the model drug nevirapine (NVP). Therefore, we chose to present the data in fold induction, ie, NVP treated vs non-treated cells. This allowed to establish a valid comparison between data obtained for each model system (2D and 3D) since the mean basal values vary between models and it could in turn lead to biased conclusions. Nevertheless, those values were calculated from the individual values obtained for each condition as shown below (new Supplementary figure 1):
Supplementary Figure 1. Phase I and phase II enzymatic induction in 3D- and 2D-HLCs after 3 (D27) and 10 days (D34) of NVP treatment. The data for hnMSC, cultured in monolayer and exposed to NVP for 3 days, are presented as negative control. All results are presented as the mean ± SD of at least three independent experiments.
Comment 6: need to explain some abbreviation (eg. 5-AZA)
Response:
We thank the Reviewer for the highlight. The abbreviation was substituted by the full name, 5-Azacitidine.
Reviewer 3:
Comment 7: The authors report that their objective was to evaluate the usefulness of a human based 3D in vitro stem cell derived-hepatic model for long-term drug metabolism and bioactivation. The authors propose that LC 3D Cultures are more Efficient in Maintaining the Dynamics of Glutathione Pools. The authors report that the glutathione flux profile of 3D cultures is altered upon NVP exposure in a time-dependent manner characterized by a reduction in net intracellular glutathione availability, a higher excretion of glutathione and a change in glutathione dynamics negatively affecting GSSP pools.
This is a well-written, well-organized article. My overall impression is that the authors rigorously utilized the bioactivation profile of one of drug with a known toxicological profile in order to evaluate their rigorously formulated 3D model with an accepted 2D model. Because only one drug was evaluated the overall importance of the manuscript is somewhat reduced. In order to increase the enthusiasm of this well-written article, the Introduction should provide more about differences between 2D and 3D models instead of focusing only on what is unknown. I would have more enthusiasm if the authors had evaluated several classes of drugs, or more drugs in this particular class. I believe that these authors are capable of evaluating more drugs which would support the generalizability of the entire Discussion. Additionally, from the data presented, one could argue that the difference between the 3D vs the 2D model is a matter of days in culture. Even so, the authors may use this finding to provide guidance for future researchers in this area. All of these points need to be addressed in the Discussion. The focus on glutathione flux is interesting; however, need to provide in-depth comments regarding other measurements of metabolic flux and why these measurements are not applied in this model. These latter suggested comments will help guide the reader to their conclusion.
Response:
We thank the Reviewer comments. Indeed, the evaluation of several classes of drugs would be very interesting and should be explored in future studies. Herein, however, our objective was to perform a demonstrative in-depth study of a known model drug whose biotransformation involves phase I, II and III metabolism. This itself is already a complex issue and the inclusion of other drugs could disperse the focus.
Moreover, a new paragraph was now included on the Discussion to cover the mentioned discussion points (now the second paragraph of the discussion) as follows:
“Several 3D in vitro liver models have been reported in the literature over the past years [5,21,23,29–34]. However, a comprehensive and systematic comparison between distinct cell culture systems that would allow its wide adoption for pharmacological and toxicological applications is scarce, whereas data from stem cell derived HLCs is even scarcer. Importantly, most hepatic differentiation protocols do not generate fully mature hepatocytes with respect to a diversity of mature hepatic functions, including drug metabolizing capacity, albumin production, urea cycle activity, or glycogen storage ability. In addition, the time in culture is frequently overlooked, despite the reports supporting that cells need time in culture to mature and to re-establish key ultrastructural and physiological traits [35]. In fact, the comparison of cell culture systems and cell sources is a challenging task. Cell viability and stability, morphological and architectural features, phase I and phase II metabolic capacity, response to a large panel of well-accepted reference drugs, as well as the physiological preservation of key metabolic and signalling pathways under long-term hepatic cultures shall be taken into consideration for pharmacological and toxicological studies. Altogether, these topics highlight the need for the proper definition of a set of consensus criteria, comprising critical elements such as cell viability, morphology, functionality and toxicological characterization, as well as time in culture [3]. It further supports that the choice of reference drugs for showing the relevance of the cell culture model and of the cell source is neither trivial nor consensual.”
The authors also acknowledge the Reviewer’s comment regarding the metabolic flux measurements. The focus of this work was to address glutathione dynamics in a more integrated approach, rather than to perform the classic measurement of the reduced/oxidized glutathione ratio. In this context, the metabolites and pools of glutathione metabolism, have been measured as an indirect of evaluating the presence and activity of the enzymes responsible for their formation. Finally, we would like to highlight that this study is, to the best of our knowledge, the first approach towards the glutathiolomic analysis (or glutathione metabolomics) in human HLCs and, therefore, the data presented here would support a variety of future studies.
Comment 8: Line 36–38, remove the words “better and” in the following sentence since the goal of these models is reliability : The well-known drawbacks of primary hepatocytes and hepatic cell lines regarding long-term metabolic stability and competence has prompted efforts for the development of better and more reliable in vitro hepatotoxicity models.
Response:
I thank the Reviewer’s highlight. The words “better and” have been deleted.
Comment 9: Lines 60–61, improve transition: In addition, the study of glutathione metabolism, in the context of drug toxicity, has not been yet explored in HLC models. This sentence can be skipped or moved to the prior paragraph without changing overall content.
Response:
This sentence has been deleted from the Introduction and moved to the Discussion Section (Line 325-326) in order to highlight this innovative aspect of this study.
Comment 10: Line 109: Figure 2C. Please present the data in the same order as the legend.
Response:
The legend from Figure 2 has been changed accordingly and now it reads as follows:
“Figure 2. Gene expression analyses of Phase I, II and III related genes of 3D- and 2D-HLCs after 3 (D27) and 10 days (D34) of NVP treatment. (A) Gene expression of Phase I enzymes comprising CYP3A4, CYP2B6 and CYP2D6 in NVP-treated cells relative to non-treated cells. (B) Gene expression of Phase II enzymes comprising SULT1A1, UGT1A1 and GSTA1-A2 in NVP-treated cells relative to non-treated cells. (C) Gene expression of the MRP7 transporter in NVP-treated cells relative to non-treated cells. (A, B and C) Data for hnMSC exposed to NVP for 3 days and used as negative controls, are also shown. (D) Phase I, II and III gene expression in 3D-HLCs relative to 2D-HLCs in NVP-treated cells at D27 and D34. The data (Mean ± SD, n = 3) are normalized to the reference gene β-ACTIN and expressed as the log10 of the ratios.”
Reviewer 2 Report
Cipriano et al. present a well established model and reasonable results about in vitro hepatic metabolism of anti-HIV drug, nevirapine. Overall structure and results are well written and interesting. I have the following concerns for the authors to consider regarding this work.
1. Nevirapine is metabolized via CYP3A4/3A5, CYP2B6, CYP2D6, and CYP2C9 as authors mentioned. Nevirapine induces CYP2B6 and CYP3A4, but CYP2D6 is not inducible form in general condition. because CYP2C9 and CYP2D6 is quite important in general drug metabolism as major isoforms of CYP and in pharmacogenetics. Thus, the differences of 2-OH, 3-OH, 8-OH, 12-OH can be related with difference expression level of those CYP isoforms (as the referenced paper, ERICKSON et al. 1999). But, present research and former research paper from same research group did not show other CYPs activity. samples from two time points preparation is limited explain the consquence of enzyme induction including CYPs, SULTs, UGTs.
2. ECOD (courmain 7-hydroxylation) is more representative for CYP2A6 in human generally. (Pelkonen O, Rautio A, Raunio H, Pasanen M. CYP2A6: a human coumarin 7-hydroxylase. Toxicology. 2000 Apr 3;144(1-3):139-47.) authors mentioned ECOD represent CYP2B6 and CYP1A1/1A2, this is not appropriate substrate and metabolites for CYP2B6, 7-benzyloxy resorufin O-dealkylation or bupropion hydroxylation is better reaction for CYP2B6. thus induction of phase I enzyme (figure 3) is not enough to explain the metabolism of phase I metabolism of nevirapine. as same as gene expression level, CYP2D6 activity is required.
3. The concept of 'glutathiolomic' is very nice, but the consequence of glutathione catabolism in more dominants in kidney than liver via Gamma-glutamyltransferase (GGT). For the explain of glutathione metabolism, glutathione conjugate of nevirapine (eg.NVP-12-glutathione), cysteinyl/mercapturic acid conjugate metabolites levels and gamma-glutamylcysteine ligase, GGT mrp2 and mrp5 expression/activity are also required.
4. It is better showing activities of CYPs in (eg. molar concentration per protein) in D27, D34 and control group, to show how comparable to standard procedure (primary hepatocytes)
Minor comments:
1. need to explain some abbreviation (eg. 5-AZA)
Author Response

(The authors gave the same response as above.)

Reviewer 3 Report
The authors report that their objective was to evaluate the usefulness of a human based 3D in vitro stem cell derived-hepatic model for long-term drug metabolism and bioactivation. The authors propose that LC 3D Cultures are more Efficient in Maintaining the Dynamics of Glutathione Pools. The authors report that the glutathione flux profile of 3D cultures is altered upon NVP exposure in a time-dependent manner characterized by a reduction in net intracellular glutathione availability, a higher excretion of glutathione and a change in glutathione dynamics negatively affecting GSSP pools.
Overall comments:
This is a well-written, well-organized article. My overall impression is that the authors rigorously utilized the bioactivation profile of one of drug with a known toxicological profile in order to evaluate their rigorously formulated 3D model with an accepted 2D model. Because only one drug was evaluated the overall importance of the manuscript is somewhat reduced. In order to increase the enthusiasm of this well-written article, the Introduction should provide more about differences between 2D and 3D models instead of focusing only on what is unknown. I would have more enthusiasm if the authors had evaluated several classes of drugs, or more drugs in this particular class. I believe that these authors are capable of evaluating more drugs which would support the generalizability of the entire Discussion. Additionally, from the data presented, one could argue that the difference between the 3D vs the 2D model is a matter of days in culture. Even so, the authors may use this finding to provide guidance for future researchers in this area. All of these points need to be addressed in the Discussion. The focus on glutathione flux is interesting; however, need to provide in-depth comments regarding other measurements of metabolic flux and why these measurements are not applied in this model. These latter suggested comments will help guide the reader to their conclusion.
Specific comments:
Introduction:
Line 36 – 38, remove the words “better and” in the following sentence since the goal of these models is reliability : The well-known drawbacks of primary hepatocytes and hepatic cell lines regarding long-term metabolic stability and competence has prompted efforts for the development of better and more reliable in vitro hepatotoxicity models.
Lines 60 – 61, improve transition: In addition, the study of glutathione metabolism, in the context of drug toxicity, has not been yet explored in HLC models. This sentence can be skipped or moved to the prior paragraph without changing overall content.
Results:
Line 109: Figure 2 C. Please present the data in the same order as the legend.
Discussion:
Kindly see overall comments
Methods:
Kindly see overall comments.
Author Response

(The authors gave the same response as above.)

Round 2
Reviewer 2 Report
Thank you to authors for appropriate revision. I would like to suggest accept revised version of the paper.